# Effects of Cabya (*Piper retrofractum* Vahl.) Fruit Developmental Stage on VOCs

**DOI:** 10.3390/foods11162528

**Published:** 2022-08-21

**Authors:** Jue Wang, Rui Fan, Yiming Zhong, Hongli Luo, Chaoyun Hao

**Affiliations:** 1Spice and Beverage Research Institute, Chinese Academy of Tropical Agricultural Science (CATAS), Wanning 571533, China; 2College of Tropical Crops, Hainan University, Haikou 570228, China; 3Key Laboratory of Genetic Resources Utilization of Spice and Beverage Crops, Ministry of Agriculture, Wanning 571533, China; 4Key Laboratory of Processing Suitability and Quality Control of the Special Tropical Crops of Hainan Province, Wanning 571533, China

**Keywords:** cabya (*Piper retrofractum* Vahl.), VOCs, GC-MS, PCA, HCA

## Abstract

The differences in VOCs can affect the flavor and medicinal value of cabya, and the flavor changes that occur in stages as the fruit develops are currently unknown. In order to investigate the influence of the developmental stage on the aroma composition of cabya essential oil, VOCs at each of the four developmental stages were analyzed by steam distillation (SD) extraction combined with GC-MS detection. The similarities and differences in fruit composition among the developmental stages were evaluated using hierarchical cluster analysis (HCA) and principal component analysis (PCA). A total of 60 VOCs, mainly alcohols, alkenes and alkanes, were identified across all of the developmental stages. The most acidic substances were detected in phase A and have a high medicinal value. There was no significant difference between the B and C phases, and the alcohols in those phases mainly promoted terpenoid synthesis in the D phase. Constituents during the D phase were mainly alkenes, at 57.14%, which contributed significantly to the aroma of the essential oil. PCA and HCA both were able to effectively differentiate the cabya fruit developmental stages based on the SD-GC-MS data. In summary, this study investigated the flavor variation characteristics and the diversity of VOCs in cabya fruits at different developmental stages, and its findings can provide a reference for developing essential oil products for different uses and determining appropriate stages for harvesting cabya resources.

## 1. Introduction

Cabya (*Piper retrofractum* Vahl), a species of pepper [1], is a woody climber in the Piperaceae family native to Southeast Asia, and mostly cultivated in Indonesia and Thailand [2]. As the “king of spices”, pepper has been widely used as a food flavoring and folk medicine for thousands of years [3]. Piperonal, an important aromatic component of black pepper, is a simple aromatic aldehyde with a cherry-like aroma that has wide utility in the flavor and fragrance industry [4]. β-caryophyllene, germacrene D, limonene, β-pinene, and other target odor components have also contributed to pepper as a valuable spice and food-flavoring product [5]. In Indonesian folk medicine, the fruits of cabya are used as a tonic in the treatment of various digestive, stimulant, hypnotic, and intestinal disorders [2]; this pepper variety is also traditionally used to treat stomach pain, rheumatoid arthritis, diarrhea, and other general infections, mainly because of its biologically active compounds [6]. The use of bioactive compounds from medicinal plants as therapeutic agents has been an important area of biomedical and natural product research, and cabya in particular has been shown to have potential nutritional and phytochemical values that supports its medicinal function as a traditional anti-flatulence, expectorant, sedative, and anti-irritant therapeutic [7,8].

The pepper species, such as cabya, are important oil-bearing commercial crops in the tropics, and the characteristic aroma and therapeutic activity of cabya is very useful in the food industry and herbal medicine [9]. Despite this, few international studies have been conducted on the essential oil of cabya. The current literature has focused on the effects of different processing methods on the extracts and the physicochemical properties of the fruit [10]; experiments have also been conducted to determine the biological activity and synthesis of extracts [11]. LC Hawa performed a study [12] examining the moisture content of cabya at three different maturation stages, represented by green, orange, and red coloration, along with the dried piperine, antioxidants, and reducing sugars. Jungon Yun et al. [7] investigated the anti-photoaging effects of standardized P. retrofractum extracton UVB damage to human dermal fibroblasts and the skin of hairless mice. T Subsuebwong et al. [13] extracted fresh cabya fruits by hydro-distillation and assayed the insecticidal activity in what was the first report of P. retrofractum essential oil having an adulticidal activity against mosquito vectors. In support of developing the essential oil products for different applications and to clarify appropriate harvesting stages, the present work extracted and characterized the essential oil from cabya fruits at different stages of development.

Specifically, the essential oil extraction was performed by steam distillation (SD), and GC-MS detection was employed to identify the aroma components. The VOCs in the different stages were comprehensively analyzed and compared by hierarchical clustering and principal component analysis. It was expected that cabya fruits at different developmental stages differ significantly in terms of aroma components, and that the representative components at different periods have different roles and effects, providing data support for the aroma characteristics of cabya that can be used to inform the selection of fruit harvesting periods for processing and production.

## 2. Materials and Methods

### 2.1. Materials

The experimental material consisted of the fresh fruit of cabya in good growth condition, grown in the germplasm nursery of the Institute of Spices and Beverage Research, Chinese Academy of Tropical Agricultural Sciences. The fruit development was classified into four periods, denoted as A, B, C, and D (Figure 1). The picked fruits were processed by blanching at 70 °C for 30 s. After the blanching treatment, the fruits were cooled and then placed in a rotary blast drying oven for drying at a temperature of 40 °C and a time of 16 h.

### 2.2. Steam Distillation (SD)

Steam distillation was conducted according to GB/T 17527-2009, the determination of the essential oil content of pepper for essential oil extraction, with slight modifications of the process conditions. First, the dried cabya fruit were crushed and passed through a 40-mesh sieve. The pepper samples (25.00 g) were then accurately weighed into a 1000 mL round-bottom flask and a small amount of explosion-proof glass beads added, followed by 250 mL distilled water (for a ratio of 1:10 pepper: water). The essential oil extraction device was then assembled, the mixture boiled for the indicated time, and distillation carried out for 4 h under continuous boiling (about five drops per minute dripped from the condenser). Subsequently, the heat source was turned off and the solution allowed to cool before reading the volume of the extracted essential oil, which was accurate to 0.05 mL.

### 2.3. GC-MS Analysis

The GC conditions were as follows: HP-5MS column (30 m × 0.25 mm × 0.25 μm); injection mode splitless; injection port temperature 250 °C; injection volume 0.5 μL; solvent delay 4 min; carrier gas helium (99.999% purity); and flow rate 1 mL/min. The temperature was programmed as follows: initial column temperature 40 °C; maintained for 5 min; increased to 220 °C at a rate of 4 °C/min; maintained for 5 min; increased to 300 °C at a rate of 10 °C/min; and finally held for 5 min.

The MS conditions were as follows: electron impact energy 70 eV; ion source temperature 230 °C; transmission line temperature 300 °C; and scan range 30–400 amu.

### 2.4. Identification of VOCs

The NIST spectral library 14 search and retention index qualitative determination was used to identify the volatile components of the cabya essential oil. The area normalization approach was used to calculate the relative content of each of the chemicals. The formula for calculating the linear retention index (LRI), where the number of carbon atoms in n-alkanes is C7–C30, is as follows:$$LRI=100\times (\frac{t-t_{n}}{t_{n+1}-t_{n}})$$

The formula reads as follows: *n* and *n*+1 are the number of carbon atoms of n-alkane before and after the component to be measured, respectively, *t_n_* and *t_n_*_+1_ are the peak retention time of the corresponding n-paraffin, t is the peak retention time of the component to be measured, where *t_n_* < *t* < *t_n_*_+1_.

### 2.5. Statistical Analysis

The experiments were repeated three times, and the data were presented in the form of mean ± standard deviation; in order to verify the significance of variation among the samples, ANOVA was used to detect the differences between the batches (*p* < 0.05). Excel 2019 was used to integrate and summarize the GC-MS data with regard to the detected VOCs. Origin 2022 (Origin Lab Inc., Northampton, MA, USA) was used to perform the principal component analysis (PCA) and to compile the aroma component category statistics from the GC-MS data. On-line software was used to generate the Venn diagram of aroma components (https://hiplot.com.cn/ (accessed on 2 May 2022)). TBtools was used to perform the hierarchical cluster analysis (HCA) on the GC-MS data.

## 3. Results and Discussion

### 3.1. SD-GC-MS Analysis of VOCs

The volatility profile of the cabya essential oil constitutes a very important sensory characteristic for the differentiation and application of the cabya fruits at different developmental stages. In order to assess the flavor variation of the cabya essential oil, the VOCs were detected, using SD-GC-MS to provide the volatility distribution of each sample at each of the four harvesting stages. This yielded a total of 60 different VOCs, which were classified into six groups according to their chemical properties: alcohols (12); phenols (4); acids (5); alkanes (34); alkenes (28); and esters (9) (Table 1).

The main active components in pepper essential oil are terpenes, which contribute to the formation of its aroma [5,14]. As presented in Table 1, alkenes are more abundant during the D phase, in which the common VOCs were β-ocimene, α-copaene, caryophyllene, β-copaene, germacrene d, β-bisabolene, and α-panasinsanene. In a previous study by Farré-Armengol et al. [15], β-ocimene was observed to be a key plant volatile with multiple relevant functions, depending on the organ and the time of emission. Syn Kok Yeo et al. [16] showed theβ-bisabolene isomer to be the main component of essential oil obtained from the medicinally valuable opoponax (Commiphora guidottii), and also suggested that β-bisabolene could be further studied for its application in the treatment of breast cancer. In the meantime, the major aroma constituent of -copaene, which has been identified in the fruits of Rubus ulmifolius [17], Criollo cacao [18], and Ocimum tenuiflorum leaves [19], is a complex, highly volatile, and extensively distributed plant compound [20].

When comparing the different developmental stages of cabya, no significant difference was observed between the periods B and C in terms of the number and content of substances detected, which were mainly alcohols, alkenes, and alkanes. Two particular alcohols, nerolidol and apitol, were only present in these periods. These structural isomers can be converted into many sesquiterpenes by non-enzymatic acid-catalyzed reactions [21], and thereby contributed to the observed increase in alkenes in phase D and the aroma of ripe fruits. It is important to note that the most prevalent component of the B phase, -eudesmol, has been found to be an appropriate raw material for natural products with anti-disease, anti-Alzheimer’s, and anti-spasmodic qualities [22]. Junenol and selina-6-en-4-ol, were the only alcohols detected in phase D. Junenol is found in many members of the Asteraceae family and is reported to have several pharmacological activities [23], while selina-6-en-4-ol has been identified by Eloisa Helena A. Andrade et al. [24] and Sarin Tadtong et al. [25] as the main chemical component of Citrus aurantium essential oil. Since ancient times, the essential oils of aromatic plants have been used for medicinal purposes, included in pharmaceutical, biomedical, cosmetic, food, veterinary, and agricultural applications [26].

The results revealed certain differences in the composition of the cabya fruit essential oil at different periods, which is a normal phenomenon and conforms to the law of secondary metabolism. Each phase features its own unique ingredients, which have a certain medicinal effect and have a high value in production and medicinal use. The differences and diversity of the VOCs at the different developmental stages of cabya fruit are listed in Table 1.

### 3.2. Comparison of VOCs by Category

To gain a more detailed understanding of the VOCs in the cabya samples, the 60 identified components were classified into six categories by type and their distributions visualized using a stacked column chart and Venn diagram (Figure 2a,b).

Figure 2a illustrates the total relative content and different types of compounds in the essential oils from each stage. In total, about (17,21,23), and VOCs were identified in the A, B, C, and D fruit development periods, respectively. The alcohols and alkenes gradually increased in number as the fruit ripened, peaking in phase D. Interestingly, the acids were highest in phase A and then gradually decreased, and were not detected in periods C and D (Figure 2a). Thus, the cabya fruits at different developmental stages may exhibit significantly different types and amounts of aroma compounds, which may partly depend on their growth phase. During phase A, in addition to achieving their highest concentrations, the acids were among the most abundant volatile compounds. Of the four acids identified, the unsaturated fatty acid 9-octadecenoic acid and the monosaturated fatty acid oleic acid were the main VOCs. Previously, Mohammad Zahid et al. [27] detected 9-octadecenoic acid using GC-MS in Annona squamosa L; moreover, the extract from the seeds of that plant was reported to have high antioxidant activity. It is important to note that oleic acid, a monounsaturated substance that is plentiful in olive oil and almonds, is utilized to produce energy, cosmetics, food goods, and medications, which have significant nutritional and therapeutic significance [28]. Thus, our findings imply that the cabya fruits in phase A may have strong antioxidant capacity and medicinal value.

The Venn diagram in Figure 2b summarizes the differences in the aroma components of cabya fruit at different periods. In general, the alkenes are key aroma compounds for the flavor of black pepper essential oil. Here, the alkenes were the most characteristic components of the essential oil from cabya fruit in phase D, mainly caryophyllene (27.71%), α-humulene (12.85%), and germacrene D (12.36%); all three were previously reported by Orav et al. to be primary VOCs in the essential oil of Piper nigrum [29]. Caryophyllene is the main sesquiterpene hydrocarbon found in black pepper essential oil [30], and the main component of the pepper aroma. It is widely used in medicine, food, cosmetics, and other industries [31]. It is also currently considered suitable for the construction of natural product-like compounds through diversity-directed synthesis; using natural products as a starting point in this process is an emerging strategy for constructing diverse backbones to meet the need for high-throughput screening in drug development [32]. As part of the pepper scent, the sesquiterpenes, such as α-humulene and germacrene D, which make up the biggest subgroup of terpenes, also have biological properties that are beneficial to health [33,34]. α-humulene has specifically been shown to have anti-inflammatory and anti-cancer activities, with great potential for medical applications [35]. Germacrene D has a wide range of insecticidal activity and the ability to act as a precursor for many other sesquiterpenes, with very good prospects for product production [36]. It is also a phyto-pheromone with roles in the biological interactions between different species [37]. Given the abundance of these major alkenes, we hypothesize that the cabya fruits in phase D feature the main VOCs of pepper essential oil, and so may be suitable for the production of pepper essential oil.

### 3.3. Principal Component Analysis (PCA) of Aroma Components

Principal component analysis (PCA) is commonly used to analyze diverse data, particularly for the purposes of dimensionality reduction and visualization [38]. Here, the 60 VOCs detected in the pepper samples at the four developmental periods were formed into a 4 × 60 matrix for principal component analysis. The eigenvalues and loading matrix resulting from the PCA are given in Table 2 and Table 3, respectively, while the score diagram and loading diagram are shown in Figure 3.

The results showed that the contribution rates of PC1, PC2, and PC3 were 61.63%, 24.93%, and 7.41%, respectively. The cumulative contribution of the first three principal components was 87.93%, which indicates these components are able to fully reflect the main information of the original variables and also achieve the purpose of this analysis. The loading matrix revealed that most volatile substances, i.e., apitol, α-eudesmol, and germacrene D, were highly positively correlated with, and the main contributors to, PC1. More broadly, the loadings on the first principal component show the alkenes and alcohols as the components having the highest correlation with and being the main characteristic VOCs of the cabya aroma. Germacrene D is known to be widely present in pepper essential oil, and hence the main VOC of pepper [5]. Regarding PC2, the main contribution came from linalool, which had a high positive correlation with PC2.

Plotting PC1 × PC2 gave a possible sample distribution based on the post-harvest aroma composition of the cabya fruits at different developmental stages (Figure 3a). In agreement with the HCA results, the samples clustered into two groups: those from periods A, B, and C located at the upper right of the plot, while phase D segregated to the bottom left. Figure 3b plots the relationship of the compound type with the first two principal components. The volatile substances, phenols, alkanes, and alcohols were highly positively correlated with PC1, and the alkenes with PC2. In summary, the results show that PCA can be used to distinguish the key VOCs of cabya fruit at different developmental stages, which can guide the subsequent development of different potent essential oils.

### 3.4. Hierarchical Clustering Analysis (HCA) of Key Aroma Components

In order to more intuitively visualize the differences between the volatile flavor components of cabya at each of the four developmental periods, we constructed a clustering heat map (Figure 4).

This new data matrix consisted of 60 VOCs and their corresponding relative contents to create a matrix of size 4 samples and 60 variables to be used for the HCA (Figure 4). The cabya samples were clearly divided into three groups, with those from periods B and C having a high material similarity and belonging to the same group. At the subsequent level of clustering, phase A and the group consisting of periods B and C grouped together, however the cabya samples from phase D were significantly different from those from all of the other periods. This classification was consistent with the PCA results, and all of the cabya samples were distinguished according to their aroma composition. While there was a certain visual difference in the fruit color among the A, B, and C periods, the Euclidean distance between the corresponding cluster groups was small and the mathematical distinction between them was not obvious. This may be because the hierarchical clustering highlights the differences in the types of compounds, while the principal component analysis encapsulates the overall flavor differences.

## 4. Conclusions

Pepper essential oil is often used as an ingredient in cosmetics, phytotherapy, aromatherapy, nutritional products, perfumes, and fragrances. The present study analyzed the variation in the cabya fruit aroma components across four developmental stages by GC-MS. A total of 60 aroma components were detected and classified into six categories, based on functional groups: 12 alcohols; 4 phenols; 5 acids; 34 alkanes; 28 alkenes; and 9 esters. Overall, the VOCs of the cabya fruit were found to be broadly similar across the developmental periods, and the alcohols and olefins were identified as the characteristic components of cabya. Some differences between the stages were evident from the patterns in the hierarchical clustering heatmap, which were confirmed by subsequent principal component analysis; the discrimination results were verified to prove that they were also diverse. Phase D exhibited a significant difference from the other three periods, and the main unique components were alkenes and alcohols, with the alkenes primarily sesquiterpenes, such as caryophyllene, α-humulene, germacrene D, and copaene. Notably, the sesquiterpenes make major contributions to fragrance and also have anti-inflammatory and anti-cancer activities, with prospects for use in medical biotherapy and the commercial applications of fragrances. The Phase B and C components were not significantly different, and those components are likely to mainly promote the formation of terpenoids in phase D, in addition to being suitable for pharmacological treatment in their own rights. Meanwhile, phase A exhibited the greatest abundance of acidic substances, and the volatiles identified in that phase feature high antioxidant activity.

The results showed that there were differences in the essential oil components in the six periods of cabya, and the secondary metabolic substances in the fruit changed due to the different developmental stages. The results of this study provide some reference information for the rational use of cabya resources and the optimal periods for harvesting cabya fruits so as to develop the essential oils for different purposes. The analysis of the effects of the different substances was a guide for the subsequent development of different essential oil products for different purposes. Further research on the extraction and functional application of the bioactive components of wild relatives in the family Piperaceae will be conducted in the future.

## Figures and Tables

**Figure 1 foods-11-02528-f001:**
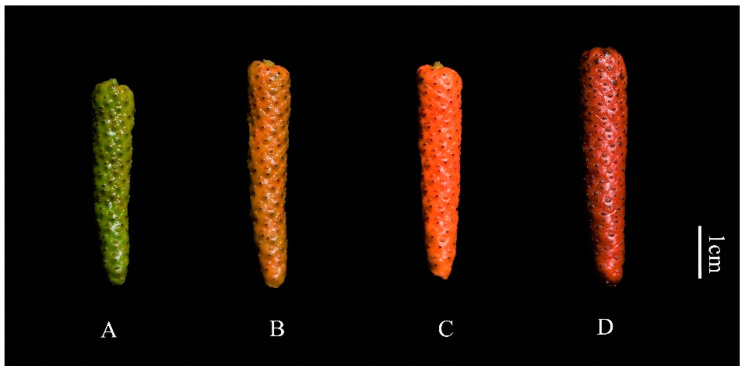
Physical phenotypes of the four developmental stages of cabya. Note: The (**A**) phase is the green ripening phase; the (**B**) phase is the color-breaking phase; the (**C**) phase is the orange phase; the (**D**) phase is the red ripening phase.

**Figure 2 foods-11-02528-f002:**
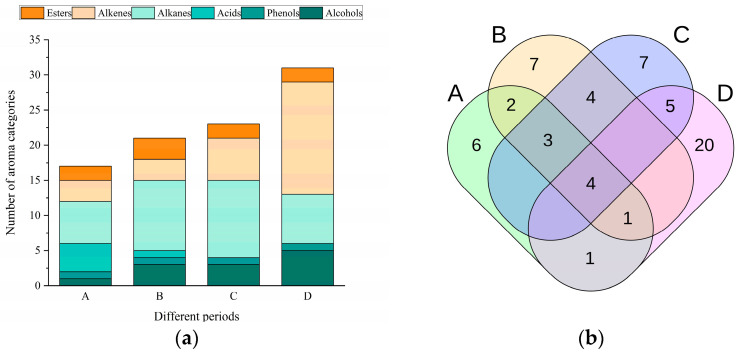
(**a**) Classification of VOCs detected in cabya fruit at each of the four developmental periods; (**b**) Venn diagram summarizing cabya fruit aroma component differences across developmental periods.

**Figure 3 foods-11-02528-f003:**
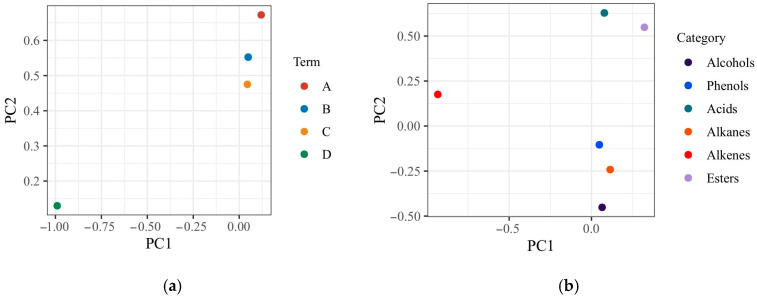
Principal component analysis (PCA) of 60 VOCs detected in cabya fruit during four periods of development. (**a**) Scatter plot of PCA scores for each phase; (**b**) PCA loading diagram for the types of VOCs.

**Figure 4 foods-11-02528-f004:**
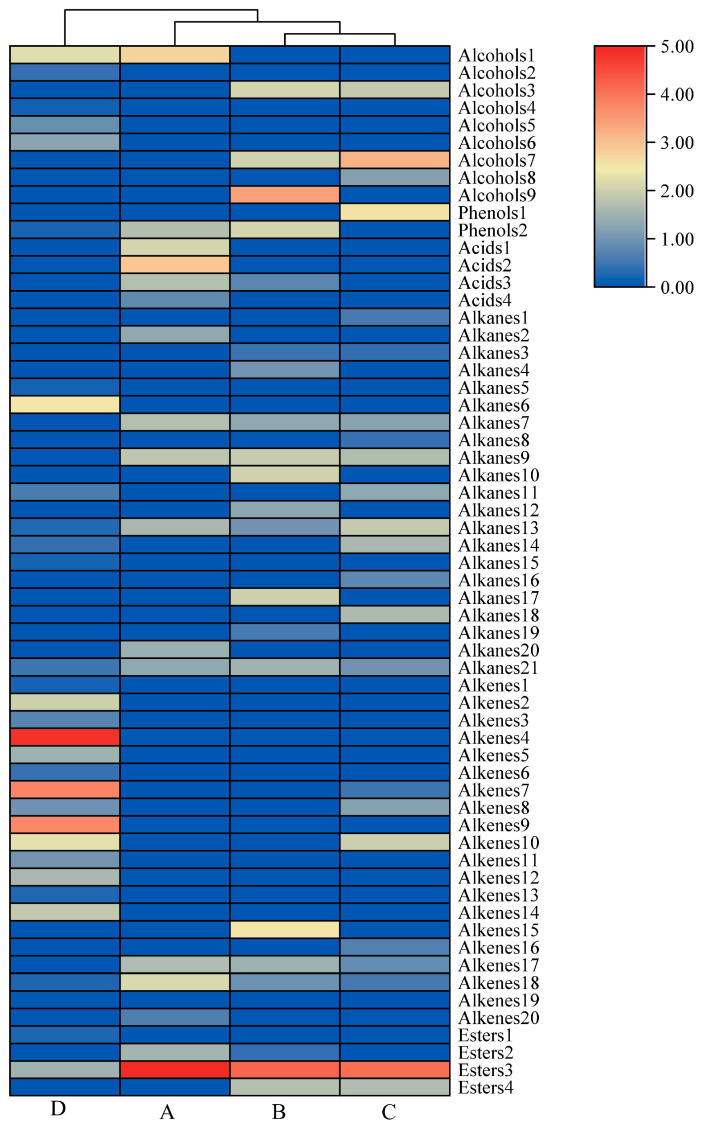
Hierarchical cluster analysis (HCA) of the four cabya fruit color stages based on content of 60 VOCs.

**Table 1 foods-11-02528-t001:** Main aromatic components of cabya essential oil and their relative contents at each stage.

No.	Category	LRI	Type ID	Component	Relative Percentage/%			
					A	B	C	D
1	Alcohols	903.46	Alcohols1	Linalool	5.73 ± 0.02 ^a^		−	3.75 ± 0.00 ^b^
2		1078.70	Alcohols2	Safrole	−	−	−	0.33 ± 0.00 ^a^
3		1253.80	Alcohols3	Nerolidol	−	3.26 ± 0.01 ^a^	2.72 ± 0.01 ^b^	−
4		1280.56	Alcohols4	3,3-Dimethyl-2-(3-methyl-1,3-butadienyl)-cyclohexane-1-methanol	−	−	−	0.13 ± 0.01 ^a^
5		1286.68	Alcohols5	Junenol	−	−	−	0.88 ± 0.01 ^a^
6		1291.38	Alcohols6	Selina-6-en-4-ol	−	−	−	1.40 ± 0.01 ^a^
7		1297.94	Alcohols7	Apitol	−	3.15 ± 0.02 ^b^	8.12 ± 0.01 ^a^	−
8		1308.36	Alcohols8	Ledol	−	−	1.29 ± 0.01 ^a^	−
9		1310.22	Alcohols9	α-Eudesmol	−	9.77 ± 0.02 ^a^	−	−
10	Phenols	1158.34	Phenols1	Methyl eugenol	−	−	4.94 ± 0.01 ^a^	−
11		1408.44	Phenols2	2,2′-Methylenebis [6-(1,1-dimethylethyl)-4-methyl-phenol	2.29 ± 0.04 ^b^	3.30 ± 0.01 ^a^	−	0.15 ± 0.01 ^c^
12	Acids	1326.79	Acids1	9-Octadecenoic acid	3.27 ± 0.04 ^a^	−	−	−
13		1338.63	Acids2	Oleic acid	6.52 ± 0.02 ^a^	−	−	−
14		1424.48	Acids3	Non-ahexacontanoic acid	2.28 ± 0.05 ^a^	0.71 ± 0.02 ^b^	−	−
15		1435.20	Acids4	1H-1,2,3-Triazolo [4,5-c] quinoline-1-hexanoic acid	0.72 ± 0.04 ^a^	−	−	−
16	Alkanes	853.41	Alkanes1	4-Ethyldecane	−	−	0.49 ± 0.01 ^a^	−
17		1058.01	Alkanes2	8-Methylheptadecane	1.50 ± 0.01 ^a^	−	−	−
18		1058.21	Alkanes3	5-Methyl-5-propylnonane	−	0.40 ± 0.01 ^a^	0.32 ± 0.01 ^b^	−
19		1063.32	Alkanes4	2,3,6-Trimethyldecane	−	1.03 ± 0.00 ^a^	−	−
20		1065.77	Alkanes5	1-Iodotetradecane	−	−	−	0.13 ± 0.01 ^a^
21		1078.06	Alkanes6	(1α,2β,4β)-1-Vinyl-1-methyl-2,4-bis(1-methylvinyl) cyclohexane	−	−	−	4.77 ± 0.02 ^a^
22		1136.39	Alkanes7	1-Iodooctadecane	2.35 ± 0.05 ^a^	1.50 ± 0.01 ^b^	1.38 ± 0.01 ^c^	−
23		1199.79	Alkanes8	2-Methylpentadecane	−	−	0.34 ± 0.01 ^a^	−
24		1242.06	Alkanes9	2,6,10,14-Tetramethylpentadecane	2.59 ± 0.03 ^b^	2.82 ± 0.01 ^a^	2.25 ± 0.01 ^c^	−
25		1291.83	Alkanes10	9-Octylheptadecane	−	3.14 ± 0.01 ^a^	−	−
26		1300.05	Alkanes11	1-Iodoeicosane	−	−	1.49 ± 0.01 ^a^	0.53 ± 0.01 ^b^
27		1308.13	Alkanes12	1-Iodopolytriacontane	−	1.46 ± 0.01 ^a^	−	−
28		1320.80	Alkanes13	2,6,10,14-Tetramethylhexadecane	2.04 ± 0.01 ^b^	1.00 ± 0.01 ^c^	2.74 ± 0.00 ^a^	0.22 ± 0.00 ^d^
29		1322.86	Alkanes14	2-Methylhexacosane	−	−	2.01 ± 0.01 ^a^	0.32 ± 0.02 ^b^
30		1335.30	Alkanes15	1,7,11-Trimethyl-4-(1-methylethyl) cyclotetradecane	−	−	−	0.16 ± 0.01 ^a^
31		1340.27	Alkanes16	3-Methylheptadecane	−	−	0.75 ± 0.01 ^a^	−
32		1365.80	Alkanes17	1-Iododocosane	−	3.05 ± 0.01 ^a^	−	−
33		1368.28	Alkanes18	Heptacosane	−	−	2.14 ± 0.01 ^a^	−
34		1387.76	Alkanes19	2,6,10-Trimethyltetradecane	−	0.51 ± 0.01 ^a^	−	−
35		1434.51	Alkanes20	7,9-Dimethylhexadecane	1.73 ± 0.02 ^a^	−	−	−
36		1404.43	Alkanes21	3-Methyloctadecane	1.48 ± 0.01 ^b^	1.85 ± 0.01^a^	0.99 ± 0.01 ^c^	0.41 ± 0.01 ^d^
37	Alkenes	848.77	Alkenes1	β-Ocimene	−	−	−	0.15 ± 0.01 ^a^
38		1098.55	Alkenes2	4-Vinyl-4-methyl-3-(1-methylvinyl)-1-(1-methylethyl)-cyclohexene	−	−	−	3.05 ± 0.01 ^a^
39		1123.89	Alkenes3	α-Copaene	−	−	−	0.67 ± 0.01 ^a^
40		1156.37	Alkenes4	Caryophyllene	−	−	−	27.7 ± 0.01 ^a^
41		1161.42	Alkenes5	β-Copaene	−	−	−	1.84 ± 0.01 ^a^
42		1171.95	Alkenes6	(s,1z,6z)-8-Isopropyl-1-methyl-5-methylenecyclodec-1,6-diene	−	−	−	0.34 ± 0.01 ^a^
43		1178.71	Alkenes7	α-Humulene	−	−	0.41 ± 0.01 ^b^	12.84 ± 0.02 ^a^
44		1191.89	Alkenes8	(+)-δ-Cadinene	−	−	1.34 ± 0.00 ^a^	0.95 ± 0.02 ^b^
45		1196.53	Alkenes9	Germacrene D	−	−	-	12.35 ± 0.01 ^a^
46		1199.58	Alkenes10	Aromandendrene	−	−	3.02 ± 0.01^b^	3.94 ± 0.01 ^a^
47		1212.23	Alkenes11	β-Bisabolene	−	−	−	1.00 ± 0.01 ^a^
48		1220.23	Alkenes12	α-Panasinsanene	−	−	−	2.00 ± 0.01 ^a^
49		1233.38	Alkenes13	4-[(1e)-1,5-Dimethyl-1,4-hexadien-1-yl]-1-methyl-cyclohexene	−	−	−	0.18 ± 0.02 ^a^
50		1303.38	Alkenes14	Copaene	−	−	−	2.76 ± 0.01 ^a^
51		1305.45	Alkenes15	β-Guaiene	−	4.74 ± 0.00 ^a^	−	−
52		1380.30	Alkenes16	1-Eicosene	−	−	0.61 ± 0.01 ^a^	−
53		1385.27	Alkenes17	1-Octadecene	2.23 ± 0.02 ^a^	1.80 ± 0.01 ^b^	0.83 ± 0.01 ^c^	−
54		1375.32	Alkenes18	1-Hexacosene	3.47 ± 0.02 ^a^	0.95 ± 0.01 ^b^	0.48 ± 0.01 ^c^	0.20 ± 0.01 ^d^
55		1460.06	Alkenes19	Z-12-Pentacosene	−	−	−	0.03 ± 0.00 ^a^
56		1428.23	Alkenes20	3-Heptadecene	0.56 ± 0.02 ^a^	−	−	-
57	Esters	1068.45	Esters1	Methyl benzoate	−	−	−	0.17 ± 0.01 ^a^
58		1392.66	Esters2	Triacontyl heptafluorobutyrate	1.91 ± 0.03 ^a^	0.32 ± 0.01 ^b^	−	−
59		1402.41	Esters3	Dibutyl phthalate	29.13 ± 0.11 ^a^	17.32 ± 0.02 ^b^	15.87 ± 0.01 ^c^	1.86 ± 0.01 ^d^
60		1411.91	Esters4	11-Tetradecen-1-ol acetate	−	2.31 ± 0.04 ^a^	2.25 ± 0.01 ^b^	−

Note: Different lowercase letters in the same column indicate significant differences between treatments (*p* < 0.05). −Indicate not detected.

**Table 2 foods-11-02528-t002:** Eigenvalues and variance contributions of the four principal components.

Principal Component	Initial Eigenvalues			Extraction Sums of Squared Loadings		
	Eigenvalue	Variance/%	Cumulative%	Eigenvalue	Variance/%	Cumulative%
1	2.47	61.63	61.63	2.47	61.63	61.63
2	1.00	24.93	86.56	1.00	24.93	86.56
3	0.30	7.41	93.98	0.30	7.41	93.98
4	0.24	6.02	100.00	0.24	6.02	100.00

**Table 3 foods-11-02528-t003:** Principal component loading matrix.

VOCs	PC1	PC2	PC3	PC4
Alcohols1	0.19	0.63	−0.01	−1.24
Alcohols2	−0.61	−0.29	−0.02	−0.10
Alcohols3	0.74	−0.34	0.06	0.86
Alcohols4	−0.61	−0.34	−0.02	−0.10
Alcohols5	−0.62	−0.16	−0.01	−0.09
Alcohols6	−0.63	−0.04	−0.01	−0.09
Alcohols7	2.00	−0.25	−1.53	1.81
Alcohols9	1.47	−0.41	2.57	1.34
Phenols1	0.57	−0.29	−1.45	0.77
Phenols2	0.43	−0.30	0.85	−0.08
Acids1	−0.12	−0.31	−0.03	−0.78
Acids2	0.36	−0.24	−0.04	−1.45
Alkanes6	−0.11	−0.33	0.16	−0.48
Alkanes9	−0.49	−0.35	−0.02	−0.26
Alkanes10	−0.49	−0.36	−0.16	−0.02
Alkanes13	−0.38	−0.34	−0.02	−0.41
Alkanes14	−0.44	−0.37	−0.01	0.01
Alkanes17	−0.39	−0.37	0.25	0.05
Alkanes18	−0.61	−0.34	−0.02	−0.10
Alkenes2	−0.68	0.76	0.02	−0.05
Alkenes4	0.39	−0.31	−0.03	−0.11
Alkenes7	−0.52	−0.36	−0.12	−0.04
Alkenes9	0.92	−0.29	0.07	0.18
Alkenes14	0.06	−0.38	0.82	0.36
Alkenes15	−0.26	−0.22	−0.45	0.17
Alkenes18	−0.29	−0.38	0.37	0.11
Esters3	0.56	−0.24	−0.55	0.11
Esters4	−0.13	−0.26	−0.60	0.26

## Data Availability

Data is contained within the article.

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
