# Peer review of "Effects of Cabya (Piper retrofractum Vahl.) Fruit Developmental Stage on VOCs"

_foods, 2022, doi:10.3390/foods11162528_

Round 1

Reviewer 1 Report

Subject of this manuscript is very interesting. The experimental design needs to be improved. Manuscript needs major improvement.

How many repetitions of the GC/MS analysis were made?

Why the results were not expressed in concentrations instead in percentages?

The authors do not mention significant or insignificant differences when presenting the results (it should be presented with addition of a standard deviation).

The authors should calculate and add in Table 1. linear retention indices (LRI) of aroma compaunds instead retention time.

Author Response

Response to Reviewer 1 Comments

Point 1: Subject of this manuscript is very interesting. The experimental design needs to be improved. Manuscript needs major improvement.

Response 1: Thank you very much for your approval of our manuscript's content. We will make detailed improvements to the experimental analysis methods and results based on your suggestions.

Point 2: How many repetitions of the GC/MS analysis were made?

Response 2: Three separate GC-MS analyses were conducted. Because the results were more consistent when the data was processed and the discrepancies were not significant, three parallel processing results were left out of this article. We are sorry for our errors and will make them right away as a result. Table 1 has been improved after processing three replicates, and added to the experimental method, on page 3, Lines 119-120.

Point 3: Why the results were not expressed in concentrations instead in percentages?

Response 3: Thanks for your opinion. The peak area of the substance in the ion chromatogram was used to calculate the GC-MS results in this experiment, and the result obtained by normalizing the peak area was a relative percentage. Because the final result we expect is the aroma components of essential oils of cabya at various developmental stages, only qualitative experiments are included in the overall experimental results. As a result, the outcomes are not given in terms of concentration.

Additionally, we reference some studies by other researchers that used peak area normalization to determine the relative concentrations of various substances.

Attached references:

[1] Hou, J., L. Liang, and Y. Wang. "Volatile composition changes in navel orange at different growth stages by HS-SPME–GC–MS." Food Research International 136(2020):109333.

[2] Asikin, Y., et al. "Volatile aroma components and MS-based electronic nose profiles of dogfruit (Pithecellobium jiringa) and stink bean (Parkia speciosa)." Journal of Advanced Research 9.C(2018):79-85.

[3] Shuai, X. , et al. "Analysis of Volatile Components in Gui Qi Mango Fruit Wine." IOP Conference Series: Earth and Environmental Science 440.2(2020):022041 (5pp).

Point 4: The authors do not mention significant or insignificant differences when presenting the results (it should be presented with addition of a standard deviation).

Response 4: Following the reviewer's advice, we corrected our work, computed the mean and standard deviation of the three replicate GC-MS data, and utilized the ANOVA approach to confirm that the samples were different (P 0.05). Additionally, add to the experimental techniques section from lines 119 to 121 on page 3. Table 1 lists the specific correction results.

Point 5: The authors should calculate and add in Table 1. linear retention indices (LRI) of aroma compaunds instead retention time.

Response 5: In accordance with the reviewer's advice, we revised this section, replaced the retention time in Table 1 with the linear retention index (LRI), and applied the calculation formula of LRI for data processing. The detailed rectification findings are displayed in Table 1.

Reviewer 2 Report

The manuscript entitled "Effects of cabya (Piper retrofractum Vahl.) fruit developmental 2

stage on volatile components” aimed at describing the differentiation of VOCs cabya fruits at four different developmental stages by hierarchical clustering and principal component analysis. I think the quality of this work should be improved  with major revisions:

Please use VOCs for volatile organic compounds through the manuscript.

The manuscript is hard to follow and extensive editing of English language and style is required. Some examples:

-Lines 128: β-ocimene was concluded to be… should be β-ocimene was observed to be…

-Lines 130: determined the β-bisabolene should be showed that β-bisabolene

-Lines 132: and also concluded that β-bisabolene should be and also suggested that β-bisabolene

-Lines 133-136: The sentence is not clear. Please, rewrite

-Line 138: in terms of the classes and quantities should be in terms of the number and content

-Lines 143-145: The sentence is not clear. Please, rewrite

-Line 145: “Two other alcohols” should be eliminated. It is a repetition in the same sentence.

-Line 147: family Asteraceae should be Asteraceae family

-Lines 150-152: “Since ancient times, aromatic plants have seen common use for medicinal purposes [26]. The essential oils of these plants are used in pharmaceutical, biomedical, cosmetic, food, veterinary, and agricultural applications” should be “Since ancient times, the essential oils of aromatic plants have been used for medicinal purposes, included in pharmaceutical, biomedical, cosmetic, food, veterinary, and agricultural applications [26]”.

-Line 155: each of should be eliminated

-Line 167: respectively should be at the end of the sentence.

-Line 173: of volatile should be volatile without of

-Lines 178-181: The sentence is not clear. Please, rewrite

-Line 185 and line 248: more characteristic should be the most characteristic

-Lines 195-197: The sentence is not clear. Please, rewrite

-Lines 205-209: the sentence is more suitable for the conclusion section.

-Lines 220-222: the sentence is not clear, please rewrite

-Lines 224-226: the sentence is not clear, please rewrite

-Line 240: of four developmental shoul be at four developmental

-Line 262: Does “All told” mean “Overall”? Please, specify

Minor points

- Line 122 and line 163: (see Table 1 for detail) should be (Table 1)

- PCA results should precede the hierarchical clustering heatmap and text should be amended accordingly

Author Response

Point 1: Please use VOCs for volatile organic compounds through the manuscript.

Response 1: All volatile organic compounds in the manuscript have been replaced with VOCs, as suggested by the reviewer.

Point 2: -Lines 128: β-ocimene was concluded to be… should be β-ocimene was observed to be…

Response 2: We have made correction according to the Reviewer’s comments. The statements of “Lines 128,β-ocimene was concluded to be” was changed to “β-ocimene was observed to be a key plant volatile with multiple relevant functions depending on the organ and the time of emission.” On page 4, Lines 142.

Point 3: -Lines 130: determined the β-bisabolene should be showed that β-bisabolene

Response 3: We have made correction according to the Reviewer’s comments. The statements of “Lines 130, determined the β-bisabolene” was changed to “showed that β-bisabolene isomer to be the main component of essential oil obtained from the medicinally valuable opoponax.” On page 4, Lines 144.

Point 4: -Lines 132: and also concluded that β-bisabolene should be and also suggested that β-bisabolene

Response 4: We have made correction according to the Reviewer’s comments. The statements of “Lines 132: and also concluded that β-bisabolene” was changed to “and also suggested that β-bisabolene could be further studied for its application in the treatment of breast cancer.” On page 4, Lines 146.

Point 5: -Lines 133-136: The sentence is not clear. Please, rewrite.

Response 5: We have rewritten this part according to the Reviewer’s suggestion. “Lines 133-136: Meanwhile, α-copaene is known to be a complex, highly volatile, and widely distributed plant compound[17]; it has been reported in Rubus ulmifolius fruits[18], Criollo ca-cao[19], and the essential oil of Ocimum tenuiflorum leaves[20], and is its main aroma component.” was changed to “In the meantime, the major aroma constituent of -copaene, which has been identified in the fruits of Rubus ulmifolius[18], Criollo cacao[19], and Ocimum tenuiflorum leaves[20], is a complex, highly volatile, and extensively distributed plant com-pound[17].” On page 4, Lines 147-150.

Point 6: -Line 138: in terms of the classes and quantities should be in terms of the number and content

Response 6: We have made correction according to the Reviewer’s comments. The statements of “Lines 138: in terms of the classes and quantities” was changed to “no significant difference was observed between periods B and C in terms of the number and content of substances detected.” On page 4, Lines 152.

Point 7: -Lines 143-145: The sentence is not clear. Please, rewrite

Response 7: We have rewritten this part according to the Reviewer’s suggestion. “Lines 143-145: It is worth mentioning that α-eudesmol, the most abundant constituent in the B period, has been reported suitable as a raw material for natural products having anti-Alzheimer's disease and anti-spasmodic properties[22]” was changed to “It's important to note that the most prevalent component of the B period, α-eudesmol, has been found to be an appropriate raw material for natural products with anti-disease Alzheimer's and anti-spasmodic qualities[22].” On page 4, Lines 157-159.

Point 8: -Line 145: “Two other alcohols” should be eliminated. It is a repetition in the same sentence.

Response 8: We have made correction according to the Reviewer’s comments. “Junenol and selina-6-en-4-ol, were the only alcohols detected in period D”, instead of “Lines145, Two other alcohols, junenol and selina-6-en-4-ol, were the only alcohols detected in period D.” On page 4, Lines 159.

Point 9:. -Line 147: family Asteraceae should be Asteraceae family

Response 9: We have made correction according to the Reviewer’s comments. “Junenol is found in many members of the Asteraceae family and is reported to have several pharmacological activities”, instead of “Lines147, Junenol is found in many members of the family Asteraceae and is reported to have several pharmacological activities.” On page 4, Lines 161.

Point 10: -Lines 150-152: “Since ancient times, aromatic plants have seen common use for medicinal purposes [26]. The essential oils of these plants are used in pharmaceutical, biomedical, cosmetic, food, veterinary, and agricultural applications” should be “Since ancient times, the essential oils of aromatic plants have been used for medicinal purposes, included in pharmaceutical, biomedical, cosmetic, food, veterinary, and agricultural applications [26]”.

Response 10: We have rewritten this part according to the Reviewer’s suggestion. “Lines 150-152: Since ancient times, aromatic plants have seen common use for medicinal purposes [26]. The essential oils of these plants are used in pharmaceutical, biomedical, cosmetic, food, veterinary, and agricultural applications” was changed to “Since ancient times, the essential oils of aromatic plants have been used for medicinal purposes, included in pharmaceutical, biomedical, cosmetic, food, veterinary, and agricultural applications [26].” On page 4, Lines 164-166.

Point 11: -Line 155: each of should be eliminated

Response 11: We have made correction according to the Reviewer’s comments. “Each period features its own unique ingredients, which has a certain medicinal effect and has high value in production and medicinal use.”, instead of “Lines 155, Each period features its own unique ingredients, each of which has a certain medicinal effect and has high value in production and medicinal use.” On page 4, Lines 169.

Point 12: -Line 167: respectively should be at the end of the sentence.

Response 12: We have made correction according to the Reviewer’s comments. “VOCs were identified in the A, B, C and D fruit development periods respectively”, instead of “Lines167, volatile compounds were respectively identified in the A, B, C and D fruit development periods.” On page 6, Lines 181.

Point 13: -Line 173: of volatile should be volatile without of

Response 13: We have made correction according to the Reviewer’s comments. “acids were among the most abundant volatile without of compounds”, instead of “Lines173, acids were among the most abundant of volatile compounds.” On page 6, Lines 188.

Point 14: -Lines 178-181: The sentence is not clear. Please, rewrite

Response 14: We have rewritten this part according to the Reviewer’s suggestion. “Lines 178-181: It is worth mentioning that oleic acid is a monounsaturated compound abundant in olive oil and almonds, and is used in the production of energy, cosmetics, nutritional products, and pharmaceuticals, in which last role it has considerable nutritional and medicinal value[29]” was changed to “It's important to note that oleic acid, a monounsaturated substance that is plentiful in olive oil and almonds, is utilized to produce energy, cosmetics, food goods, and medi-cations, which has significant nutritional and therapeutic significance[29].” On page 7, Lines 192-195.

Point 15: -Line 185 and line 248: more characteristic should be the most characteristic

Response 15: We have made correction according to the Reviewer’s comments. “Here, alkenes were the most characteristic components of essential oil from cabya fruit in period D”, instead of “Lines185, Here, alkenes were more characteristic components of essential oil from cabya fruit in period D”, and “Lines 236: The loading matrix revealed that most volatile substances”, instead of “Lines248, The loading matrix revealed that more volatile substances.” On page 7, Lines 200.

Point 16: -Lines 195-197: The sentence is not clear. Please, rewrite

Response 16: We have rewritten this part according to the Reviewer’s suggestion. “Lines 195-197: Meanwhile, α-humulene and germacrene D are also sesquiterpenes, that being the largest subgroup of terpenes, and have health-promoting bioactive effects as part of the pepper aroma[33,34]” was changed to “As part of the pepper scent, sesquiterpenes such as α-humulene and germacrene D, which make up the biggest subgroup of terpenes, also have biological properties that are beneficial to health[33,34].” On page 7, Lines 210-212.

Point 17: -Lines 205-209: the sentence is more suitable for the conclusion section.

Response 17: As suggested by the reviewer, which is true, we put “Lines 205-209: The results showed that there were differences in the essential oil components in the six periods of cabya, and the secondary metabolic substances in the fruit change due to the different developmental stages. And the analysis of the effects of different substances was a guide for the subsequent development of different essential oil products for different purposes.” in the conclusion section“The results showed that there were differences in the essential oil components in the six periods of cabya, and the secondary metabolic substances in the fruit change due to the different developmental stages. The results of this study provide some refer-ence information for the rational use of cabya resources and the optimal periods for harvesting cabya fruits so as to develop essential oils for different purposes. And the analysis of the effects of different substances was a guide for the subsequent develop-ment of different essential oil products for different purposes.” On page 10-11, Lines 300-306.

Point 18: -Lines 220-222: the sentence is not clear, please rewrite

Response 18: We have rewritten this part according to the Reviewer’s suggestion. “Lines 220-222: Sixty volatile compounds and the corresponding relative contents were the elements of this new data matrix, constructing a matrix of size 4 samples × 60 variables to perform the HCA (Figure 3)” was changed to “This new data matrix consisted of 60 VOCs and their corresponding relative contents to create a matrix of size 4 samples 60 variables to be used for the HCA (Figure 4).” On page 9, Lines 263-264.

Point 19: -Lines 224-226: the sentence is not clear, please rewrite.

Response 19: We have rewritten this part according to the Reviewer’s suggestion. “Lines 224-226: At the next level of clustering, period A and the group comprised of periods B and C clustered together, with cabya samples from period D being significantly different from all other periods” was changed to “At the subsequent level of clustering, period A and the group consisting of periods B and C grouped together, however cabya samples from period D were significantly different from those from all other periods.” On page 9, Lines 266-269.

Point 20: -Line 240: of four developmental should be at four developmental

Response 20: We have made correction according to the Reviewer’s comments. “Here, the 60 VOCs detected in pepper samples at four developmental periods”, instead of “Lines240, Here, the 60 volatile components detected in pepper samples of four developmental periods.” On page 8, Lines 228.

Point 21: -Line 262: Does “All told” mean “Overall”? Please, specify

Response 21: We are very sorry for our erroneous writing, Line 262 "All told" mean "In summary", we have made correction, “In summary, the results show that PCA can be used to distinguish the key VOCs of cabya fruit at different developmental stages” instead of “Lines 262, All told, the results show that PCA can be used to distinguish the key volatile compo-nents of cabya fruit at different developmental stages.” On page 8, Lines 249-251.

Minor points

Point 1: - Line 122 and line 163: (see Table 1 for detail) should be (Table 1)

Response 1: We have made correction according to the Reviewer’s comments. “Lines 122: (see Table 1 for detail)”, was changed to“ (Table 1)”, and “Lines 163: See Table 1 for detailed data”, was changed to“Table 1”

Point 2: - PCA results should precede the hierarchical clustering heatmap and text should be amended accordingly PCA

Response 2: We have made correction according to the Reviewer’s comments. Chart unification and results analysis for PCA and HCA have been adjusted and corrected.

Round 2

Reviewer 1 Report

Now the results are better presented and properly discussed and concluded. The authors explained the lack of some information about the calculation and presentation of the results. Overall, the manuscript is well written and could be considered for publication.

Reviewer 2 Report

Manuscript can be accepted in the present form